# Receptor Transporter Protein 4 (RTP4) in the Hypothalamus Is Involved in the Development of Antinociceptive Tolerance to Morphine

**DOI:** 10.3390/biom12101471

**Published:** 2022-10-13

**Authors:** Wakako Fujita, Hitoshi Uchida, Masashi Kawanishi, Yusuke Kuroiwa, Manabu Abe, Kenji Sakimura

**Affiliations:** 1Department of Medical Pharmacology, Nagasaki University Graduate School of Biomedical Sciences, Nagasaki 852-8523, Japan; 2Department of Cellular Neurobiology, Brain Research Institute, Niigata University, Niigata 951-8585, Japan; 3Department of System Pathology for Neurological Disorders, Brain Research Institute, Niigata University, Niigata 951-8585, Japan; 4Department of Therapeutic Innovation and Pharmacology, Nagasaki University Graduate School of Biomedical Sciences, Nagasaki 852-8521, Japan; 5Department of Animal Model Development, Brain Research Institute, Niigata University, Niigata 951-8585, Japan

**Keywords:** mu-opioid receptor, receptor transporter protein 4, paraventricular nucleus, tolerance, toll-like receptor 4, mitogen-activated protein kinase

## Abstract

Receptor transporter protein 4 (RTP4), one of the receptor chaperone proteins, contributes to the maturation and membrane trafficking of opioid receptor heteromers consisting of mu (MOPr) and delta (DOPr) opioid receptors (MOPr-DOPr). Although MOPr-DOPr is known to mediate the development of morphine tolerance, the extent to which RTP4 plays a role in this process has not been elucidated. Given that RTP4 can be upregulated by repeated administration of morphine, especially in the hypothalamus, here we investigated the effect of hypothalamus-selective ablation of RTP4 on the development of antinociceptive tolerance to morphine. In this study, we generated *RTP4^flox^* mice and selectively knocked-out RTP4 using local injection of adeno-associated virus expressing Cre recombinase (AAV-Cre) into the hypothalamus. The AAV-Cre injection partially, but significantly, decreased the level of RTP4 expression, and suppressed the development of antinociceptive tolerance to morphine. Next, we examined the mechanism of regulation of RTP4 and found that, in neuronal cells, *Rtp4* induction is via Gi and MAPK activation, while, in microglial cells, the induction is via Toll-like receptor 4. Together, these studies highlight the role of MOR activity in regulating RTP4, which, in turn, plays an important role in modulating morphine effects in vivo.

## 1. Introduction

Receptor transporter protein (RTP) is a family of receptor chaperone proteins that consists of four members (RTP1, RTP2, RTP3 and RTP4) that will transport G protein-coupled receptors to the cell surface membrane [1]. RTP1 and 2 were first identified for their role in promoting cell-surface expression of odorant receptors (Saito et al., 2004), while RTP3 and 4 have been shown to increase surface expression of taste receptors in heterologous cells [2]. In addition, we and others have reported that RTP4 has an important role as an opioid receptor chaperone. For instance, it mediates the maturation and trafficking of a heteromer consisting of a mu opioid receptor (MOPr) and delta opioid receptor (DOPr) (called an MOPr-DOPr heteromer) [3]. However, the physiological and pathophysiological roles of the RTP4 chaperone remain to be assessed. Interestingly, the analysis of *Rtp* expression in mouse brain [4] reveals that RTP4 is the most abundant subtype of the RTP family in the brain. In addition, the mouse-brain database [5] suggests that *Rtp4* is broadly expressed in neurons of various brain regions, including the hypothalamus, as well as in glia, including microglia, astrocyte, Schwann cells, satellite glia and enteric glia, creating an expectation of their role in brain function.

Prolonged or repeated MOPr activation is well known to lead to the development of antinociceptive tolerance to morphine [6]. A previous study has shown that chronic morphine treatment upregulates MOPr-DOPr expression within various brain regions, including the hypothalamus [7]. The disruption of the MOPr-DOPr heteromer by a TAT-peptide also suppresses morphine tolerance [8], suggesting the possible involvement of the MOPr-DOPr heteromer. Recently, we have demonstrated that MOPr stimulation upregulates the gene expression of *Rtp4*, a chaperone of the MOPr-DOPr heteromer, in both in vitro and in vivo systems [9]. More importantly, chronic morphine treatment-induced RTP4 mRNA upregulation occurs in a hypothalamus-specific manner [9]. Thus, in this study we directly tested the extent to which hypothalamic RTP4 plays in the development of morphine tolerance using conditional deletion of RTP4 using *Rtp4*-floxed (*Rtp4^flox^*) mice.

The hypothalamus consists of several nuclei, including the paraventricular nucleus (PVN), and all opioid receptor subtypes (MOPr, DOPr and KOPr) are expressed within these areas. The present study focused on the PVN region, since it plays an important role in the descending inhibitory system (i.e., PVN-induced descending antinociception) [10]. Although the descending pathway has been considered to contribute to the antinociceptive tolerance to morphine [11], the role of PVN has not been elucidated yet. The modulation of the neural circuit, including the PVN and ventral tegmental area, has been shown to inhibit the antinociceptive tolerance to morphine [12], supporting the hypothesis of the possible involvement of PVN in the mechanism. In the present study, we targeted the PVN to locally decrease the level of RTP4 using adeno-associated viral (AAV)-mediated Cre-loxP recombination to achieve PVN-specific ablation of RTP4.

We find that chronic morphine treatment-induced morphine tolerance was diminished in PVN-specific RTP4 conditional knockout (cKO) mice. We also demonstrated that Gi activation and the MAPK pathway was involved in the mechanism of *Rtp4* gene induction after MOPr stimulation, by assessing the neuronal Neuro2A (N2A) cell line as our previous study [9]. In addition, microglial cell line SIM-A9 cells also showed morphine-induced *Rtp4* upregulation, albeit requiring a higher concentration of morphine and Toll-like receptor 4 (TLR4) but not MOPr.

## 2. Materials and Methods

### 2.1. Chemicals

The following reagents were used: DAMGO ([D-Ala2, N-Me-Phe4, Gly5-ol]-Enkephalin acetate salt) (Sigma-Aldrich, Saint Louis, MO, USA); morphine hydrochloride (Takeda Pharmaceutical, Tokyo, Japan); pertussis toxin (PTX), an inhibitor of the G protein (Gi, Go and Gt) heterotrimer interaction with receptors (Tocris, Bio-Techne Japan, Tokyo, Japan); Actinomycin D (Act D), a transcriptional inhibitor (FUJIFILM Wako Pure Chemical Corporation, Osaka, Japan); U0126, a potent, selective inhibitor of MEK1 and 2 (Tocris, Bio-Techne Japan, Tokyo, Japan); and Pyridone 6, a potent pan-JAK inhibitor (Tocris, Bio-Techne Japan, Tokyo, Japan). Morphine was dissolved in saline for animal injection. For in vitro experiments, DAMGO, morphine and PTX were diluted with H_2_O, while the other reagents were diluted with dimethyl sulfoxide.

### 2.2. Animals

Male C57BL/6N mice (20–40 g, 12–24 weeks old) were purchased from CLEA Japan, Inc., and maintained on a 12-h light/dark cycle with rodent chow and water available ad libitum. *Rtp4^flox^* mice were generated, as described below, and male mutant mice (20–40 g, 12–24 weeks old) were used. All mice were maintained on a 12-h light/dark cycle with rodent chow and water available ad libitum, and they were housed in groups of four until testing. Animal studies were carried out according to protocols approved by the Nagasaki University Animal Care and Use Committee (approval number: 2002181596) and Institutional Review Board of Niigata University (approval number: SA00041).

### 2.3. Generation of Rtp4^flox^ Mice

To generate mice carrying floxed *Rtp4* alleles (*Rtp4^flox^*), we constructed a gene-targeting vector containing loxP sites flanking exon 2 of the *Rtp4* gene and FRT-flanked neomycin (Neo) selection cassette. The plasmid vector was linearized and introduced into RENKA embryonic stem cells derived from the C57BL/6N strain and grown on feeder cells [13]. Correctly targeted clones, validated by using Southern blot analysis, were injected into ICR blastocysts and transferred to pseudo-pregnant ICR females. The male chimeras were then crossed with female C57BL/6N mice. To excise the FRT-flanked Neo cassette, the F1 offspring with successful germline transmission of the targeted allele were crossed with Actb-FLP knock-in C57BL/6N mice [14]. The resultant heterozygous *Rtp4^flox/+^* mice were intercrossed to obtain homozygous *Rtp4^flox/flox^* C57BL/6N mice. Genotype was determined by using forward (CTAGCCTGGGAAGTTAAACCCTCG) and reverse (CCGGGTACATGTGGCACAAGATCA) PCR primers. To ablate the *Rtp4* gene in a PVN-specific manner, an AAV vector expressing CMV-driven Cre recombinase, fused to eGFP (Serotype: AAV9: Addgene.org/105545, Watertown, MA, USA) (AAV-Cre) or eGFP alone (Serotype: AAV9: Addgene.org/105530) (AAV-eGFP), as a control, was bilaterally injected into the PVN, as described below.

### 2.4. Stereotactic Surgery

For intracranial injections, mice (12–24 weeks old) were deeply anesthetized with a combination anesthetic (0.75 mg/kg medetomidine, 4.0 mg/kg midazolam and 5.0 mg/kg butorphanol) before surgery. The scalp was shaved, Povidone-Iodine (Isodine^®^, Mundipharma K.K., Tokyo, Japan) was applied, and an incision was made along the midline of the scalp. The skull was scraped clean of periosteum. A micro drill (Tamiya, Inc., Shizuoka, Japan) was used to create a hole approximately 0.4 mm bilateral and 0.7 mm caudal to the bregma (bilateral PVN). A 300-nL aliquot of virus particle (2.85 × 10^9^ genome copies/side) was given per hemisphere at 400-nL/min using a syringe pump (KDS-310PLUS) (kdScientific, Holliston, MA, USA). The needle was left to stand for 10 min and then removed within 2 min. The incision was closed with a soft silk suture and antibacterial ointment was applied to the wound. The animals were allowed to recover on a disposable heating pad until they recovered by using 0.75 mg/kg of atipamezole and were returned to their home cages in the animal facility for 4 weeks until the measurement. Cannula placement into the bilateral PVN was verified with trypan blue (4%). According to the previous research related to the in vivo virus genome introduction, the viral genome has been reported to remain episomal in the nucleus but maintains sustained expression in terminally differentiated cells for several weeks to months [15,16]. To obtain stable expression of Cre-recombinase by Cre-AAV injection, we decided to perform the evaluations in this study 4 weeks after injection of the AAV-Cre-eGFP or AAV-eGFP into the PVN.

### 2.5. Drug Administration

Morphine hydrochloride was dissolved in saline. Mice were treated with morphine hydrochloride (10 mg/kg) subcutaneously once a day for 11 days.

### 2.6. Tail-Flick Test

Four weeks after the AAV injection, morphine-induced antinociception was evaluated by using the tail-flick test [6]. Using a tail-flick apparatus (IITC Life Science, Woodland Hills, CA, USA), the intensity of the heat source was set at 16–18, which resulted in the basal tail-flick latency occurring between 2 and 3 s for most of the animals. The tail-flick latency was recorded before (0 min, baseline latency) and at every 30 min after morphine injection (30, 60, 90, and 120 min); %MPE was calculated for each mouse at each time point on the indicated days (Days 4, 8 and 11) of morphine treatment according to the following formula: %MPE = ((latency after drug − baseline latency)/(10 − baseline latency)) × 100. Cutoff latency was selected at 10 s to minimize tissue damage. The area under the %MPE vs. time curves (area under the curve (AUC)) for each treatment condition is shown in Figure 2C.

### 2.7. Immunohistochemistry

On the day following the last morphine injection (i.e., 12th day), as described above, mice were deeply anesthetized with a combination anesthetic (0.75 mg/kg medetomidine, 4.0 mg/kg midazolam, and 5.0 mg/kg butorphanol) before surgery and transcardially perfused 2 min with ice-cold PBS and then for 8 min with ice-cold 4% paraformaldehyde (Wako, Osaka, Japan) for 8 min at a rate of 5–6 mL/min. Brains were extracted, post-fixed in 4% PFA at 4 °C overnight, and transferred to 10% (2 h), 20% (24 h) and then 30% sucrose in PBS for 24 h at 4 °C. Brains were frozen in O.C.T. compound (SAKURA, Tokyo, Japan) and stored at −80 °C. For virus expression and RTP4 expression verification, 20–25 µm coronal cryo-sections were cut using a cryostat (Leica CM3050) (Leica biosystems, Tokyo, Japan), dried and stored at −80 °C until use. For immunostaining, brain sections were gently washed 3 × 5 min in PBS and blocked with blocking solution (PBS containing 3% bovine serum albumin and 0.3% Triton X-100) for 1 h at room temperature (RT). Sections were then incubated in rabbit anti-RTP4 antibody (1:200) (MyBioSource, San Diego, CA, USA) in blocking solution at 4 °C overnight. Sections were rinsed 3 × 5 min with PBS and incubated secondary antibody (1:500) (Alexa-594-conjugated Goat anti-rabbit IgG) (Life Technologies Corp, Carlsbad, CA, USA) in blocking solution for 2 h at RT. Sections were washed 3 × 5 min with PBS, mounted on slides, and coverslipped with ProLong^TM^ Diamond Antifade Mountant with DAPI (Thermo Fischer Scientific, Waltham, MA, USA). They were stored at −30 °C until analysis.

### 2.8. Imaging and Quantification by ImageJ Software

The histochemical analysis of the stained brain sections (from 2.7.) were performed on images acquired using a confocal laser scanning microscope (LSM-800) (Carl Zeiss Meditec Co., Ltd., Tokyo, Japan) (40 × objective). The signal intensity of RTP4 in the eGFP-co-localized area (cells) was calculated using macros in Image J software.

### 2.9. N2A^MOPr^ Cell Culture and Transfection

In vitro experiments were performed as described previously [9]. Briefly, Neuro 2A (N2A) cells were grown in complete growth medium (E-MEM with 10% FBS and 1% P/S). Cells were transfected with Flag-MOPr (N2A^MOPr^ cells) using Lipofectamine 2000 according to the manufacturer’s protocol (Thermo Fisher Scientific Inc., Waltham, MA, USA). Twenty-four hours after transfection, cells were seeded into 24-well plates for further experiments.

### 2.10. Treatment of N2A^MOPr^ Cells

The experiments were performed as described previously [9]. Briefly, 24 h after plasmid transfection, N2A^MOPr^ cells were seeded into a 24-well plate (2 × 10^5^ cells/ well) in complete growth medium and incubated overnight at 37 °C with 5% CO_2_. Next day, media were replaced with 500 µL of the medium containing DAMGO or vehicle at the final concentration of 10 µM and incubated for 24 h. In one set of experiments, the cells were pretreated with inhibitors or vehicle at the final concentration of 100 ng/mL (PTX), 300 pM (ActD), 10 µM (U0126), 300 nM (Cal C) or 10 pM (Pyridone 6) for 24 h (PTX) or 30 min (others) followed by co-treatment with DAMGO and inhibitors for 24 h, as indicated in the figure legends. After ligand treatment, N2A^MOPr^ cells were washed once with cold phosphate-buffered saline (PBS) and then collected with 300 µL of cold RLT buffer (QIAGEN, Hilden, Germany). The cell lysates were transferred into an RNase-free centrifuge tube and stored at −80 °C until they were further processed for RT-qPCR.

### 2.11. SIM-A9 Cell Culture and Treatment

SIM-A9 cells (ATCC, Manassas, VA, USA) were grown in complete growth medium (D-MEM with 10% FBS, 5% HS, and 1% P/S) at 37 °C with a 5% CO_2_ condition. SIM-A9 cells (8 × 10^4^) in culture medium were seeded into 24-well plates one day before the experiment. The next day, media were replaced with 500 µL of the complete growth medium containing morphine or vehicle at a final concentration of 10 µM to 1 mM and incubated for 3 to 48 h at 37 °C with 5% CO_2_. In one set of experiments, cells were pretreated with inhibitors or vehicle at a final concentration of 1 to 100 µM (naltrexone; NTX) or 1 to 10 µg/mL of TLR4/MD-2 monoclonal antibody (Thermo Fischer Scientific, Waltham, MA, USA) as indicated in the figures for 30 min followed by co-treatment with morphine and reagents for 24 h. After ligand treatment, SIM-A9 cells were collected the same as N2A cells.

### 2.12. RT-qPCR

RT-qPCR was performed as described previously [9]. We extracted RNA-containing aqueous solution using the TRIzol reagent according to the manufacturer’s protocol (Thermo Fisher Scientific Inc.) before the total RNA purification. Total RNA was purified using the RNeasy Mini kit (QIAGEN Inc., Germantown, MD, USA) according to the manufacturer’s protocol. cDNA was synthesized using the PrimeScript^TM^ RT Master Mix (Takara, Shiga, Japan) according to the manufacturer’s protocol. Real-time PCR was performed using the Power SYBR Green qPCR Master Mix (Applied Biosystems, Foster City, CA, USA). The PCR template source was 4 µL of 10-times-diluted first-strand cDNA. Amplification was performed with an ABI PRISM 7900HT sequence-detection system (Applied Biosystems) or StepOne Real Time PCR system (Applied Biosystems). After an initial denaturation step at 95 °C for 10 min, amplification was performed using 45 cycles of denaturation (95 °C for 15 s), annealing (55 °C for 30 s) and extension (72 °C for 30 s). We amplified GAPDH, a housekeeping gene, as a control. The data were analyzed using the sequence-detection system software (version 2.2.1, for ABI PRISM 7900HT Software Applied Biosystems or version 2.3, for StepOne Software, Applied Biosystems) as described in Data Analysis. The software generates the baseline subtracted amplification plot of the normalized reporter values (ΔRn) versus cycle number. The amplification threshold was set at 6–7 of the ΔRn linear dynamic range (50–60% of maximum ΔRn). The fractional cycle at which the intersection of amplification threshold and the plot occurs is defined as the threshold cycle (Ct-value) for the plot. Samples that gave a Ct-value within 45 cycles were considered to be positive for mRNA expression. Then, quantitative analysis was performed using the ΔΔCT method, as described previously [9,17].

The forward (F) and reverse (R) primers were as follows:

GAPDH-F; TGAAGGTCGGTGTGAACG

GAPDH-R; CAATCTCCACTTTGCCACTG

RTP4-F; GGAGCCTGCATTTGGATAAG

RTP4-R; GCAGCATCTGGAACACTGG

TLR4-F; GCCTTTCAGGGAATTAAGCTCC

TLR4-R; AGATCAACCGATGGACGTGTAA

OPRM1-F; TCGCCTCCAACATCAGTTAG

OPRM1-R; TTTAGGGGATTGCCTTGATC

OPRD1-F; TCCCCATAACACAAATGCTG

OPRD1-R; CCGCCTTGAGATAACATCG

### 2.13. Data and Statistical Analysis

The data are expressed as the mean ± SEM. Student’s *t*-test or one-way ANOVA with a multiple-comparison test (Dunnett’s test or Tukey’s test) were used to analyze the data. A difference was considered to be significant at *p* < 0.05. All behavioral experiments were randomized, performed by a blinded researcher and then unblinded before statistical analysis. The results for each experiment were expressed as the mean ± SEM. The AUC was calculated for each animal with the baseline defined as zero. A normal distribution of the data was verified before performing parametric statistical analysis. Wherever appropriate, data were analyzed using two-way ANOVA, followed by Tukey’s post-hoc tests or multiple unpaired *t*-tests with significance set at *p* < 0.05. All calculations were performed using GraphPad Prism 9 software (GraphPad Software, Inc., San Diego, CA, USA) and EZR (version 1.40, Saitama Medical Center, Jichi Medical University, Saitama, Japan).

## 3. Results

### 3.1. Generation of Rtp4-floxed Mouse

To generate *Rtp4*-floxed mice, we constructed a targeting vector, in which the exon 2 of the *Rtp4* gene was flanked by two *loxP* sequences (Figure 1A). Exon 2 was targeted so as to delete the transmembrane domain that leads to loss-of-function on the interaction with membrane proteins, including receptors. A recombinant C57BL/6N ES clone was confirmed by Southern blot analysis (Figure 1B) was used for the generation of chimeric mice. *Rtp4*-floxed mice were created after the removal of neomycin cassette by crossing to a transgenic strain expressing the FLP1 recombinase gene. *Rtp4^flox/flox^* mice were obtained from intercrossing *Rtp4^+/flox^* offspring (Figure 1C). *Rtp4*-floxed mice showed no gross changes and they bred normally.

### 3.2. Conditional Knockdown of RTP4 Suppressed the Development of Antinociceptive Tolerance to Morphine

First, we checked whether the introduction of the targeted allele could affect basal pain sensitivity, by assessing the tail-flick withdrawal latencies. No significant difference in tail-flick latencies was observed among wild-type, *Rtp4^+/flox^* and *Rtp4^flox/flox^* animals (Figure 2A). To test the involvement of PVN RTP4, we locally injected the AAV-Cre-eGFP (AAV-Cre) or AAV-eGFP (control AAV) to the PVN of *Rtp4^flox/flox^* mice. Successful injection was verified by trypan blue staining. Four weeks after the injection, we started the evaluation of the development of antinociceptive tolerance after chronic morphine administration, as described previously [9]. Similarly, no significant difference in tail-flick baseline withdrawal latencies were detectable in AAV-Cre-injected mice, as compared to AAV-eGFP-injected mice (Figure 2B). As for the development of antinociceptive tolerance to morphine, the results indicated that PVN-specific cKO of RTP4 partially but significantly delayed the development of antinociceptive tolerance to morphine (Figure 2C); that is, the morphine antinociception in AAV-Cre group was significantly higher than that in the control group on Day 11 of repeated administration of morphine, while on Day 4 of repeated administration of morphine, rather, the AAV-Cre group exerted significant lower levels of morphine antinociception than the control group (Figure 2C). There was a decreasing rate of antinociception (AUC/day) between Day4 and Day11, 931.4 ± 39.7 and 429.3 ± 108.4, respectively (*p* = 0.0020, unpaired *t*-test), indicating the significant slower development of antinociceptive tolerance in AAV-Cre-injected mice (Figure 2C). In addition, in AAV-Cre-injected mice, prolonged and significantly higher morphine antinociception was observed on Day 11 (Appendix A). By using immunohistochemical analysis, we confirmed that the RTP4 levels were partially but significantly downregulated in AAV-Cre-infected (i.e., eGFP-positive) cells when compared to AAV-eGFP-infected cells (Figure 2D,E). Since the purpose of this study was to reveal the importance or extent of the involvement of RTP4 in the PVN hypothalamus in antinociceptive tolerance to morphine, developed by repeated systemic administration of morphine, which could be clinically observed, we administered morphine systemically, but not by local PVN administration, as the in vivo model of antinociceptive tolerance to morphine. Furthermore, the introduction of the targeted floxed (flanked by loxP) allele per se may affect the development of antinociceptive tolerance to morphine. Thus, the floxed homozygous mice but not wild-type mice were used both in the control (i.e., AAV-eGFP injected) and conditional KO (i.e., AAV-Cre injected) group.

### 3.3. Time Course of DAMGO-Induced Rtp4 Gene Expression in N2A^MOPr^ Cells

To evaluate the extent to which *Rtp4* can be induced by MOPr activation in neuronal cells, we have established an in vitro evaluation system, as described before [9]. Briefly, in N2A^MOPr^ cells, DAMGO (10 µM), a MOPr agonist, stimulation led to a gradual and significant increment in RTP4 mRNA levels at 12 to 24 h after DAMGO stimulation (Figure 3).

### 3.4. MAPK-Mediated Rtp4 Gene Expression under MOPr Stimulation in N2A^MOPr^ Cells

To elucidate the signaling cascade underlying MOPr activation-induced *Rtp4* upregulation, the N2A^MOPr^ cells were pretreated with several inhibitors, including pertussis toxin (PTX), a Gi protein inhibitor; Actinomycin D (ActD), a transcriptional inhibitor; U0126, a MAPK inhibitor; Calphostin C (Cal C), a pan-type PKC inhibitor; and Pyridone 6, a pan-type janus kinase (JAK) inhibitor. We found that the inhibitor treatment by themselves did not significantly affect the RTP4 mRNA levels, whereas treatment with PTX, ActD or U0126 (Figure 4A–C) (but not Cal C or Pyridone 6) (Figure 4D,E) significantly attenuated the DAMGO-mediated (24 h) increases in the RTP4 mRNA levels. These results support a role for a signaling pathway involving Gi and MAPK on the induction of *Rtp4* gene after MOPr stimulation.

### 3.5. Toll-Like Receptor 4 (TLR4)-Mediated Rtp4 Gene Expression by Morphine Treatment in SIMA-9 Cells

To test whether MOPr stimulation-induced *Rtp4* upregulation could occur in microglia, we treated the SIM-A9 microglial cell line with morphine at concentrations of 10 µM (lower) to 1 mM (higher) for 24 h (Figure 5A). We found that the higher concentration of morphine (1 mM) significantly increased the RTP4 mRNA levels while there were no changes in the lower (10 and 100 µM) concentration of morphine. A significant increase in RTP4 mRNA levels was observed at 24 h after the treatment with a higher concentration of morphine (1 mM) and returned to the control levels at 48 h after the treatment with morphine (Figure 5B). Interestingly, naltrexone (NTX), a MOPr antagonist, did not affect the morphine (1 mM)-induced increase in RTP4 mRNA levels, while the TLR4-targeting antibody significantly suppressed the effect of morphine (1 mM) (Figure 5C,D). There was no significant effect on the RTP4 mRNA levels by NTX or TLR4 antibody per se without morphine (Figure 5C,D). Together, these results suggest interesting regulation of RTP4 expression by TLR4 in microglia.

## 4. Discussion

In this study, we show that a moderate decrease in RTP4 expression in the PVN has a significant inhibitory effect on morphine tolerance. In our previous report, *Rtp4* was shown to be upregulated in the hypothalamus after repeated administration of morphine [9]. Here, we determined the antinociceptive effect of morphine in locally RTP4-reduced mice in the hypothalamus that were generated by using a combination of novel *Rtp4^flox/flox^* mice and local injection of Cre-expressing AAV into the PVN region of the brain.

Interestingly, AAV-Cre injection into the PVN of *Rtp4^flox/flox^* mice partially but significantly delayed the development of antinociceptive tolerance to morphine after 11 days of morphine administration, as shown in Figure 2C. These results suggest that RTP4 in PVN cells partly contribute to the mechanism of development of antinociceptive tolerance to morphine. Since there were no differences in pain sensitivity per se against heat stimulation in the tail-flick test with or without AAV-Cre injection, the downregulation of the *Rtp4* gene via the Cre-Lox system seems to simply affect the mechanism of development of antinociceptive tolerance to morphine. On the other hand, on Day 4 of repeated administration of morphine, rather, the AAV-Cre group exerted significantly lower levels of morphine antinociception than the control group. These results suggest that downregulation of RTP4 per se may induce some “compensatory effects” that “facilitate the morphine tolerance” if it is the case that RTP4 positively contribute to facilitating the development of antinociceptive tolerance to morphine.

Although the present study did not address whether the downregulation of RTP4 as a receptor chaperone protein could affect the expression of the MOPr-DOPr heteromer in vivo, this point will be shown in the future. Interestingly, a partial (~50%) but not complete RTP4 protein knockdown in the PVN was found to give a moderate phenotype. Improvement (i.e., the complete knockout of RTP4 in local brain region) will be expected by using a higher titter of AAV-Cre or neuron-selective pAAV in the future. In this study, we measured the expression levels of RTP4 in the eGFP-positive cells by immunohistochemistry, but a more definitive and quantitative measure of protein (Western blot) or mRNA (PCR) levels would also be valuable by extracting the eGFP-positive cells, while this remains the limitation in this study. Since RTP4 was shown to be broadly expressed in whole brain [4], RTP4 in other brain regions and/or in other nuclei of the hypothalamus may also contribute to the mechanism, and thus it needs further determination.

The in vitro results in the present study suggest that at least the neuronal cells could respond to MOPr stimulation and thus upregulate *Rtp4* gene via the Gi and MAPK pathway. Note that we have previously confirmed that the DAMGO-induced upregulation of the RTP4 mRNA levels was completely suppressed by MOPr antagonist NTX [9], suggesting the involvement of MOPr in the mechanism. Regarding the transcription of the *Rtp4* gene, as RTP4 is known as one of the interferon (IFN)-stimulated genes (ISGs), whose expression is induced by the activation of the inflammatory or immune system [18], it may be mediated by the signaling molecules underlying IFN receptors, such as the JAK-STAT pathway, which leads to rapid transcriptional activation of ISGs [19,20]. However, pyridone 6, a pan-type JAK inhibitor, did not affect the DAMGO-induced RTP4 gene induction in N2A^MOPr^ cells (Figure 4E), suggesting that some distinct cellular responses were exerted under MOPr activation from those under IFNr activation.

The MAPK pathway includes a range of proteins, such as p38, ERK and JNK, involved in many faces of cellular regulation, from gene expression to cell death [21,22]. Our results showed that ERK-MAPK is involved in the mechanism of *Rtp4* induction, since U0126, which inhibits MEK1 and MEK2, and thus ERK activation, suppressed the DAMGO-induced upregulation of RTP4 mRNA levels (Figure 4C). The opioid receptor is primarily controlled by interactions with G proteins and/or beta-arrestin [23,24,25] and both pathways can activate MAPK. This study revealed Gi inhibition by PTX completely blocked the DAMGO-induced *Rtp4* induction, suggesting the importance of the Gi protein-mediated pathway for *Rtp4* gene induction, while it needs further determination on the contribution of the beta-arrestin-mediated pathway. In addition, the present results indicate the insignificant effect of inhibitors such as Cal C or Pyridone 6 on the RTP4 mRNA-inducing effect of DAMGO. However, as shown in Figure 4D,E, that the significant RTP4 mRNA-inducing effect of DAMGO also disappeared in the presence of inhibitors may suggest the weak but not statistically significant involvement of PKC or JAK in the mechanism of the RTP4 mRNA-inducing effect of DAMGO, although it needs further determination.

Not only the neuronal cell, but also the involvement of glial cell, an essential contributor to the brain function, should be addressed, and our results revealed that microglial cells may respond to morphine only at a higher dose via TLR4 but not to MOPr stimulation (Figure 5). Importantly, we confirmed that SIM-A9 cells surely possesses MOPr (OPRM1) mRNA while their expression levels are 200-times lower than that of TLR4 (Appendix A). As previously reported, microglial cells can be activated by morphine administration, although it is still controversial whether they can be stimulated via MOPr per se or not [26,27,28]. Many studies have reported that morphine will be able to bind to TLR4 and the involvement of TLR4 in response to morphine in glial cells [22], consistent with the results in the present study. Importantly, not only the neuronal cells, but also RTP4 has been shown to be highly expressed in macrophages, immunoreactive cells, according to a database [4], suggesting the role of RTP4 in “microglia”, an immune cell equivalent to macrophage in the brain. Since chronic opioids can cause microglia and astrocyte activation, and interfering with glial function has been shown to reduce tolerance [26,29,30], the involvement of crosstalk between the opioid pathway and glial or immune system may contribute to the mechanism of antinociceptive tolerance to morphine [31]. Although the database suggests that RTP4 is expressed in most of the cell types, including neurons, astrocytes, microglia and Schwann cells, as described above, it would be important to reveal the cell-type specificity for the endogenous expression of RTP4 in PVN. Future studies addressing this using primary culture or immunohistochemical analysis of the brain sections from naive and morphine-tolerant mice are needed.

An increasing number of studies indicate that RTP4 can serve as one of the important markers of diagnosis or prognosis of various diseases, such as cutaneous melanoma, breast cancer, type 2 diabetes and viral infection [18,32,33,34]. Especially since ISGs including RTP4 possesses immuno-reactive properties, such as anti-virus activity [18], it may be involved in the changes in the immune response under chronic morphine treatment [31], although it needs further determination in the future. In addition, further investigation to distinguish the cell type (i.e., neurons or glia) where *Rtp4* will be upregulated after chronic morphine treatment in vivo (i.e., in the hypothalamus) should be addressed in the future.

## 5. Conclusions

In this study, we simply demonstrated that RTP4 in the hypothalamus partly but significantly contributes to the mechanism of development of morphine tolerance after repeated administration of morphine. MOPr stimulation will lead to upregulation of *Rtp4* via MAPK pathways in neuronal cells while, in microglial cells, the induction is via TLR4.

## Figures and Tables

**Figure 1 biomolecules-12-01471-f001:**
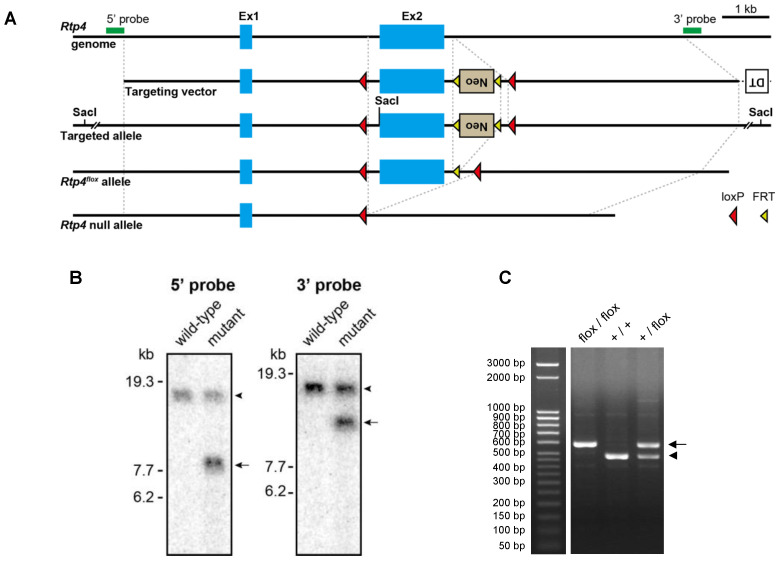
Generation of *Rtp4*-floxed mice. Schema illustrating the strategy for the generation of *Rtp4^flox^* mice and conditional KO of the *Rtp4* gene (**A**). Filled blue boxes delineate exon 1 and 2 (Ex1 and 2) of the *Rtp4* gene. Green bars indicate probes used in Southern blot analysis. Southern blot analysis of genomic DNA from wild-type mice and heterozygous mutant mice carrying the targeted allele (**B**). Hybridization with a 5′ or 3′ probe detected the 17.68-kb wild-type band (arrowhead) and 8.24- or 11.37-kb target allele bands (arrows) in Sac I-digested DNA (left and right panels). PCR genotyping of *Rtp4*-floxed mice (**C**) showing the 482-bp wild-type allele band (arrowhead) and 583-bp *Rtp4^flox^* allele band (arrow).

**Figure 2 biomolecules-12-01471-f002:**
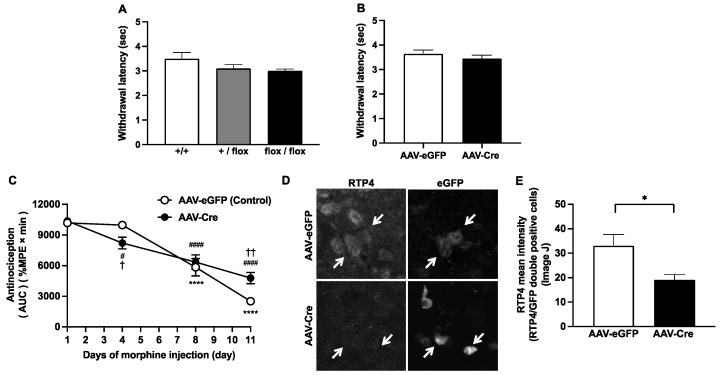
The effect of genotype on the pain sensitivity determined by tail-flick test (**A**,**B**). The influence of genotype (**A**) and the influence of AAV-Cre injection (**B**) on thermal pain sensitivity. Data are the mean ± SEM. *n* = 5–14 (**A**), *n* = 8 (**B**). +/+, wild type; +/flox, floxed hetero; flox/flox, floxed homo. The effect of knockdown of RTP4 in PVN on the development of antinociceptive tolerance to morphine (**C**). The daily changes in antinociceptive effect of morphine during repeated administration. Data are the mean ± SEM. *n* = 7 (AAV-eGFP) (control), *n* = 10 (AAV-Cre), **** *p* < 0.0001 vs. Day1 (AAV-eGFP); ^#^
*p* < 0.05, ^####^
*p* < 0.0001 vs. Day1 (AAV-Cre), Dunnett’s multiple comparison test; ^†^
*p* < 0.05, ^††^
*p* < 0.01 vs. AAV-eGFP (control), repeated measures ANOVA and the post-hoc Tukey’s multiple comparisons test. The effect of AAV-Cre on RTP4 expression in PVN area (D and E). Representative images of the immunohistochemical signal for RTP4 and the AAV-eGFP signal of the virus infected in the PVN area. Arrows indicate the eGFP-positive cell. The mean intensity of the RTP4 signal in eGFP-positive cells were measured by Image J software, as described in the Methods section (E). The mean signal intensity of RTP4 in the RTP4/eGFP double-positive cells were measured from a single cryostat brain section, including the PVN region from 7 (AAV-eGFP) and 6 (AAV-Cre) mice. Ten to 40 eGFP-positive cells were measured from each section. Data are the mean ± SEM. *n* = 7 (AAV-eGFP) (control), *n* = 6 (AAV-Cre), * *p* < 0.05 unpaired *t*-test.

**Figure 3 biomolecules-12-01471-f003:**
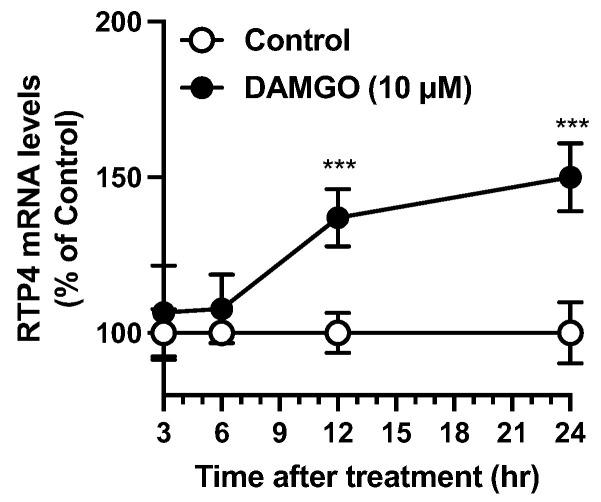
Time dependent increase of RTP4 mRNA levels after DAMGO treatment. N2A^MOPr^ cells (2 × 10^5^ cells/well) were treated with DAMGO (10 µM) at indicated periods. RT-qPCR was performed with specific primers against RTP4 and GAPDH. Data are the mean ± S.E.M. *n* = 5–6, *** *p* < 0.001 vs. control, unpaired *t*-test.

**Figure 4 biomolecules-12-01471-f004:**
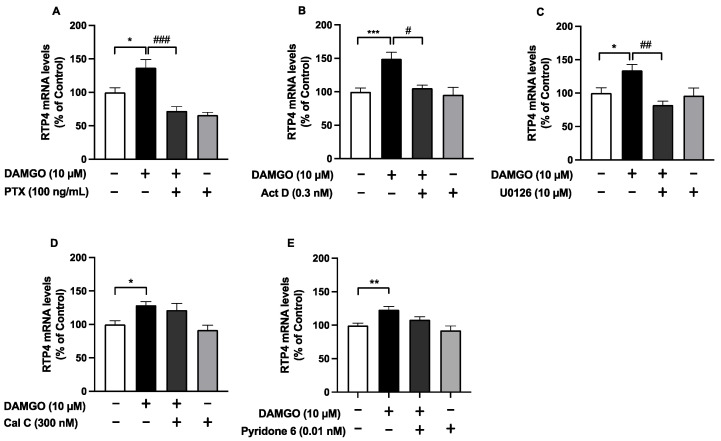
The effect of inhibitors of signal transduction molecules on DAMGO-induced (24 h) increase in RTP4 mRNA levels (**A**–**D**). N2A^MOPr^ cells (2 × 10^5^ cells/well) were treated with DAMGO (10 µM) without or with pre-treatment, with inhibitors at the indicated concentrations for 24 h (PTX) (**A**) or 30 min (ActD (**B**), U0126 (**C**), Cal C (**D**), Pyridone 6 (**E**)) before addition of DAMGO. RT-qPCR was performed with specific primers against RTP4 and GAPDH. Data are the mean ± SEM. *n* = 5–31, * *p* < 0.05, ** *p* < 0.01, *** *p* < 0.001 vs. control (open bar); ^#^
*p* < 0.05, ^##^
*p* < 0.01, ^###^
*p* < 0.001 vs. DAMGO (closed bar), Tukey’s multiple comparisons test.

**Figure 5 biomolecules-12-01471-f005:**
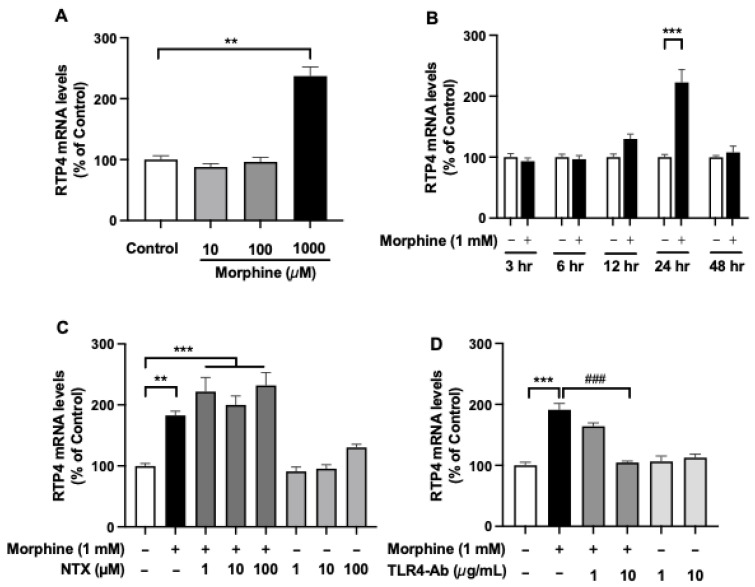
The effect of morphine on the RTP4 mRNA levels in the microglial cell line. SIM-A9 cells (8 × 10^4^ cells/well) were treated with morphine (10 µM to 1 mM) (**A**,**B**) without or with pre-treatment with MOPr antagonist (NTX) (**C**) or TLR4 antibody (TLR4-Ab) (**D**) at the indicated concentrations for 30 min before addition of DAMGO. RT-qPCR was performed with specific primers against RTP4 and GAPDH. Data are the mean ± SEM. *n* = 3–10, ** *p* < 0.01, *** *p* < 0.001 vs. control (open bar); ^###^
*p* < 0.001 vs. morphine (closed black bar), Tukey’s multiple comparisons test.

## Data Availability

Not applicable.

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
