# Peer review of "Receptor Transporter Protein 4 (RTP4) in the Hypothalamus Is Involved in the Development of Antinociceptive Tolerance to Morphine"

_biomolecules, 2022, doi:10.3390/biom12101471_

Round 1
Reviewer 1 Report
This study extends prior work indicating that at least one class of receptor transport proteins (RTP-4) is upregulated at the mRNA level following morphine treatment, that this upregulation shows some regional specificity (high in hypothalamus), and surface expression of an MOP/DOP heteromer was increased in parallel with RTP-4 expression. The current study extends in vivo relevance of RTP-4 hypothalamic expression by examining effects of hypothalamic down-regulations of this protiein in a novel line of conditional RTP-4 mice, with a focus on tolerance development While effects on tolerance are generally modest, these results are appropriately discussed by the authors. As such, this study represents an important next step in understanding RTP-4 function. Additional studies that examine signaling pathways in neuronal and microglial-like cell lines transfected with MOP provided only modest new insight that should be extended to primary neuronal and microglial cells. The following comments should be addressed:
1. The report of the RTP-4 conditional line is a novel and important aspect of this paper. Some additional clarity, though, is needed re: mating strategy of the mice used in this study. Were all conditional mice derived from homozygous matings separate from production of control mice via het/het mating? When were the control mice produced and maintained? The authors might also mention that the strain background of the Acrb—flp KI mice was identical to that of the RTP-4 conditional mice which is important and could be mentioned.
2. The difference in tolerance development between WT and conditional KO is modest and significant only at day 11 of morphine treatment in a relatively small cohorts of control and conditional mice, which is one reason that source of control mice is important to clarify since genetic heterogeneity between these groups could result is this modest effect.
3. The decrease in hypothalamic RTP-4 protein illustrated by ICC in Fig. 1d needs to be better described and perhaps expanded. How many sections were quantitated? From how many of the mice? The authors are again careful discussing this aspect of the stud but a more definitive measure of RTP-4 decrease by western/pcr in the PVN would be valuable.
4. As an intro to the4 signaling section: What is the in vivo cell-type specific expression of RTP-4 in neurons and different glial subtypes in the PVN?
5. The authors several times use “glial” cells (for example line 30) when in fact the measurements are only in a cell line representing microgila and not other glia. The authors should correct these statements.
6. The signaling studies with model cell lines modified by transfection are well-performed and convincing, but really need to be extended to primary neurons and microglia to enhance the impact of this paper.
Minor comments
Line 150—Did the authors intend to say: “on the day following the last morphine injection, day 12 ….. ?
Line 178 was confusing to this reviewer.
Author Response
Please kindly see the attachment (pdf).

Reviewer 2 Report
The work entitled “Receptor transporter protein 4 (RTP4) in hypothalamus involves in the development of antinociceptive tolerance to morphine”, provides evidence of the participation of the RTP4 receptor in the development of tolerance to morphine, despite the fact that the product of the research is adequate, I have some observations:
In lines 91 and 94, it mentions the age of the animals used in work, however, there are very noticeable differences, I consider that the authors should have better controlled this aspect to avoid variations in age, if they went from using an age range, for some experiments they used 12-15 week old mice and in other 12-24 week old mice, it is a difference in the subjects that even affects body weight.
-Although you focused on showing the evidence of the effects on the PVN, the administration of morphine was systemic, you should explain why you decided to administer it this way and not use local administration
You evaluated the effects of the introduction of targeted allele on baseline nociception, however, you show no evidence as to whether the introduction of targeted allele could change over time, i.e. why a Wild type group was not considered to rule out developmental effects of tolerance?.
Explain why the authors decided to perform the evaluations 4 weeks after injection of the AAV-Cre-eGFP (AAV-Cre) or 282 AAV-eGFP to PVN
Figure 2C shows the development of tolerance, but at 4 days there seems to be a difference between the groups evaluated, was this analysis performed? The authors should describe this point in the results.
n the statistical analysis of figures 2C and 3, they mention that they performed an unpaired t-test, I consider that it is not the most appropriate test, perhaps you should use a Repeated measures ANOVA
In the results of Figure 4D, how is it explained that the DAMGO 10 µM + Cal C 300 nM group is similar to the DAMGO 10 µM group, but also behaves the same as the control group? In figure 4E How do you explain that DAMGO 10 µM + Pyridone 6 0.01nM is not different from DAMGO 10 µM, but also there are differences with the control without any treatment?. explain in discussion
In the results shown in figure 5C, you performed a Tukey test, which compares all groups with all, in this sense, there was a statistical difference between NTX 1 µM and 10 µM vs NTX 100 µM, it seems that they are different. If you find a significant difference, how can this effect be explained?
In line 370, you mention the phrase "suppressed of the development of antinociceptive tolerance", I consider that you have to be more careful in the terminology, because it is not being suppressed, in your comparison with the basal effects you found that there is tolerance. In addition, you are reporting the data up to day 11, but it is unknown what happens after, and in line 377 you mention that on day 4, the effect is the opposite (figure S1), this data would also have to be corroborated in Figure 2C.
Line 291 mentions Figure S1, expressing that it is a figure of the supplement.
on line 92, "Ad libitum" must be written in italics
Author Response
Please kindly see the attachment (pdf).

Round 2
Reviewer 1 Report
The authors have submitted a revised manuscript that addressed essentially all my comments in the initial review. This reviewer appreciates that some additional experiments are outside the scope of the present report.
Reviewer 2 Report
The authors have responded to the suggestions and I consider that the presentation of the work has been improved, so from my point of view the work is suitable for publication.